# Real-time reliable determination of binding kinetics of DNA hybridization using a multi-channel graphene biosensor

Shicai Xu[1], Jian Zhan[2], Baoyuan Man[3], Shouzhen Jiang[3], Weiwei Yue[3], Shoubao Gao[3], Chengang Guo[1], Hanping Liu[1], Zhenhua Li[1], Jihua Wang[1] & Yaoqi Zhou[1,2]

Reliable determination of binding kinetics and affinity of DNA hybridization and single-base mismatches plays an essential role in systems biology, personalized and precision medicine. The standard tools are optical-based sensors that are difficult to operate in low cost and to miniaturize for high-throughput measurement. Biosensors based on nanowire field-effect transistors have been developed, but reliable and cost-effective fabrication remains a challenge. Here, we demonstrate that a graphene single-crystal domain patterned into multiple channels can measure time- and concentration-dependent DNA hybridization kinetics and affinity reliably and sensitively, with a detection limit of 10 pM for DNA. It can distinguish single-base mutations quantitatively in real time. An analytical model is developed to estimate probe density, efficiency of hybridization and the maximum sensor response. The results suggest a promising future for cost-effective, high-throughput screening of drug candidates, genetic variations and disease biomarkers by using an integrated, miniaturized, all-electrical multiplexed, graphene-based DNA array.

[1] Shandong Provincial Key Laboratory of Biophysics, College of Physics and Electronic Information, Dezhou University, Dezhou 253023, China. [2] Institute for Glycomics and School of Information and Communication Technology, Griffith University, Parklands Drive, Southport, Queensland 4222, Australia. [3] School of Physics and Electronics, Shandong Normal University, Jinan 250014, China. Correspondence and requests for materials should be addressed to J.W. (email: jhw25336@126.com) or to Y.Z. (email: yaoqi.zhou@griffith.edu.au).

One of the most basic experiments in molecular biology is quantitative measurement of binding kinetics and thermodynamics between two bioactive molecules such as DNA, RNA, proteins and ligands. In particular, binding interactions between small-size oligonucleotides such as DNA and RNA are of both fundamental and practical interests as they are widely used for genetic screening and detection, disease biomarkers, transcriptional profiling and single-nucleotide variant discovery[1–4]. Currently, the standard tools for binding assays are optical-based sensors such as label-free surface plasmon resonance (SPR)[5] or surface plasmon diffraction sensor[6]. However, these optical sensors are difficult to detect small changes in mass such as oligonucleotide binding[6] because optical responses depend on analyte's molecule weights. Thus, quantifying DNA hybridization with high sensitivity and precision remains a challenge by employing label-free optical sensors. Moreover, optical sensors rely on large optical components, making them difficult to reduce operational costs and to miniaturize for high-throughput measurements that are necessary for system biology studies[7] and personalized and precision medicine[1–3].

Highly promising candidates for next-generation biosensors are nanomaterial field-effect transistors (FETs)[8,9] that detect charged molecules by electronic responses. One-dimensional (1D) nanostructures such as carbon nanotubes and nanowires (NWs) have been demonstrated for real-time detection of a variety of bioactive molecules with ultrasensitivity from nM to fM level[10–18]. However, to date, few are capable of reliable, quantitative real-time kinetic measurement because of high variations in device characteristics and restriction by planar complementary metal-oxide semiconductor nanofabrication processes[19]. Recently, highly aligned carbon nanotube or NW sensors were fabricated by expensive and exquisite nanolithographic tools. These aligned NW sensors were demonstrated in quantitative measurement of protein-receptor[18] and DNA-binding kinetic constants[14]. However, the high probe density in NW sensors significantly reduces the efficiency of DNA hybridization and kinetics, with reported association constant $K_A$ that is 10–100 times smaller than that of similar length DNA measured by SPR and surface plasmon diffraction sensor[14]. To date, reliable and cost-effective nanodevices are not yet available for quantitative measurement of binding kinetics and affinity of DNA hybridization.

The recent emerging two-dimensional (2D) atomically layered materials such as graphene[20], topological insulators[21,22] and transition metal dichalcogenides[23], on the other hand, not only possess electronic, optical and mechanical properties comparable or superior to those of nanowires but, more importantly, are also compatible to planar nanofabrication processes. For example, graphene's $sp^2$-bonded honeycomb lattice exhibits ultra-high carrier mobility ($> 200,000\ cm^2\ V^{-1}\ s^{-1}$)[24] that is significantly better than FETs based on carbon nanotubes and Si nanowires[25]. Because every atom in graphene is a surface atom, it exhibits a very high surface-to-volume ratio and is ultrasensitive even to a single gas molecule[26]. By far, graphene has been employed in various biosensing platforms[8,27] for detection of glucose[28], nitric oxide[29], DNA[30–32], biomarkers[33], bacteria[34], pathogens[27] and nervous system[35]. Such a large number of studies successfully demonstrated a wide-range application of 2D biosensors with low detection limit, and a few further illustrated their capabilities in quantifying kinetics and affinity of protein-ligand binding[36–38]. However, the quantitative determination of DNA–DNA or DNA–RNA hybridization kinetics have not yet been achieved due to controllability and reproducibility across different devices and fabrication batches.

Here, we grew centimetre-scale single-crystal graphene domains and patterned them into a DNA sensor composed of six graphene FETs (G-FETs) for multiplexed analysis of DNA binding kinetics and affinities. The intrinsic 2D structure of graphene facilitates top-down fabrication of the G-FET array within a single sensor. The calibrated responses in all six FETs are well consistent with each other, enabling precise quantification of DNA concentrations as well as affinities and kinetics of DNA hybridization. These G-FET affinity sensors show high selectivity to oligonucleotides at a concentration as low as 10 pM, which is 3 orders of magnitude below the limit of detection (LOD) of current standard optical methods[28]. Moreover, the G-FET sensors were found effective in discriminating single-base mismatches in the target DNA sequence, indicating their promising future for high-throughput and reliable quantification of genetic variants and DNA biomarkers at low cost.

## Results

**Device fabrication.** DNA sensor was fabricated by using a chemical vapour deposition (CVD)-grown monolayer-dominated

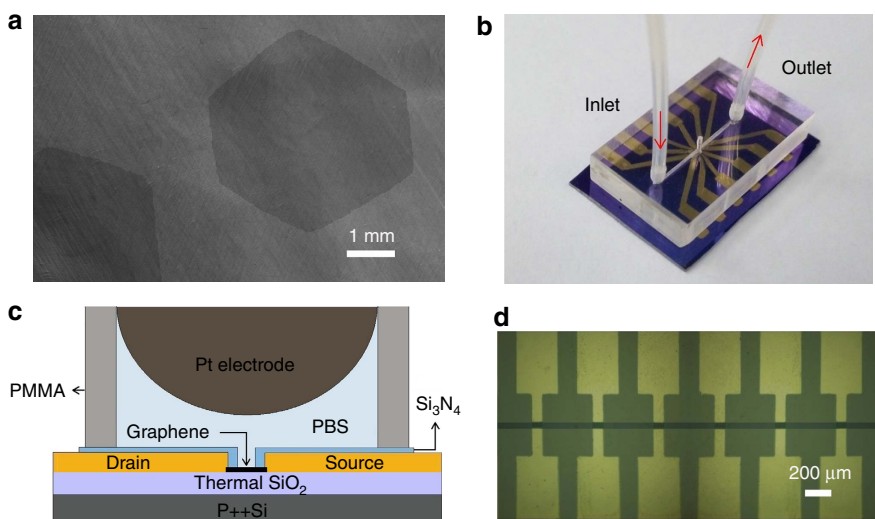

**Figure 1 | Photos and layout of the sensor chip.** (**a**) Scanning electron microscope (SEM) images of graphene single-crystal domain. (**b**) Photograph of the sensor composed of multi-G-FETs with a microfluidic channel along with solution inlet and outlet as labelled. (**c**) Cross-sectional view of an individual G-FET device. (**d**) Optical micrograph of the sensor with multi-G-FET detecting channels.

graphene single-crystal domain on a SiO$_2$/Si substrate (Supplementary Fig. 1). Here we created the sensor containing a linear array of six G-FETs, forming six parallel detecting channels. To ensure the consistency of each device, all the G-FETs were fabricated using a graphene single-crystal domain patterned by oxygen-plasma etching (Fig. 1). The source and drain contacts were formed by thermally evaporating 20 nm Cr and 100 nm Au layers onto each graphene site, keeping the remaining graphene channel at 45 μm long and 90 μm wide with a graphene channel-to-channel spacing of 400 μm. To eliminate parasitic current between metal contacts in solution, ∼80 nm of Si$_3$N$_4$ was deposited using plasma-enhanced CVD everywhere on the chip except the graphene sites and the outer pins of the Au contacts. A poly(methyl methacrylate) (PMMA) microfluidic channel was fabricated and clamped on the top of the graphene sensor array to facilitate robust and controllable reagent delivery by a home-made microfluidic system. A platinum wire was inserted into the microfluidic channel and immersed in the electrolyte to serve as the solution gate.

**Probe DNA immobilization and target DNA hybridization.**
After setting the gate voltage $V_g$ and the voltages of the source and drain of each graphene channel $V_{ds}$ to zeros, we sequentially introduced 1-pyrenebutanoic acid succinimidyl ester (PBASE) and single-stranded probe DNA. As shown schematically in Fig. 2a, PBASE binds to graphene by π stacking of its pyrene group onto the graphene surface, while the succinimide portion of PBASE extends out from the sensor surface and permits

immobilization of 5′-amine-modified probe DNAs by the conjugation reaction between the amine group of the probe DNA and the amine-reactive succinimide group of PBASE[39,40]. This process is followed by a hybridization phase when a complementary (target) or control DNA is delivered into the sensor channel (Fig. 2b). Here we employed DNA probe P20 and target T20 sequences utilized previously[41] for demonstrating kinetic measurement (Table 1). Raman spectrum records the changes of graphene in the process of PBASE binding, conjugation of the probe DNA and hybridization with the target DNA, where the characteristic peaks of PBASE and DNA[42,43] were clearly observed (Supplementary Fig. 2). The spectral changes of graphene surface demonstrated the successful immobilization of probe DNAs and their hybridizations with target DNAs. By comparison, neither PBASE nor DNA Raman signals were observed on the Pt electrode, indicating that the Pt was not functionalized in the process of graphene functionalization (Supplementary Fig. 2e). Here all six G-FETs were bound to the same probe DNA to illustrate the consistency and reliability of kinetic measurements.

Functionalization and hybridization in G-FETs can be monitored with the drain source current ($I_{ds}$) between the drain and source contacts with a constant voltage ($V_{ds} = 0.1$ V) and a varying gate voltage ($V_g$) from −1.2 to 1.7 V with the step of 50 mV s$^{-1}$ (Supplementary Fig. 3a). This one-second-step size is three orders of magnitude longer than the characteristic fall and rise times (∼0.2 ms) in response to a single pulse from 0 to 50 mV (Supplementary Fig. 3b), and thus ensures the stability and reproducibility of measured $V_g - I_{ds}$ curves and $V_{cnp}$.

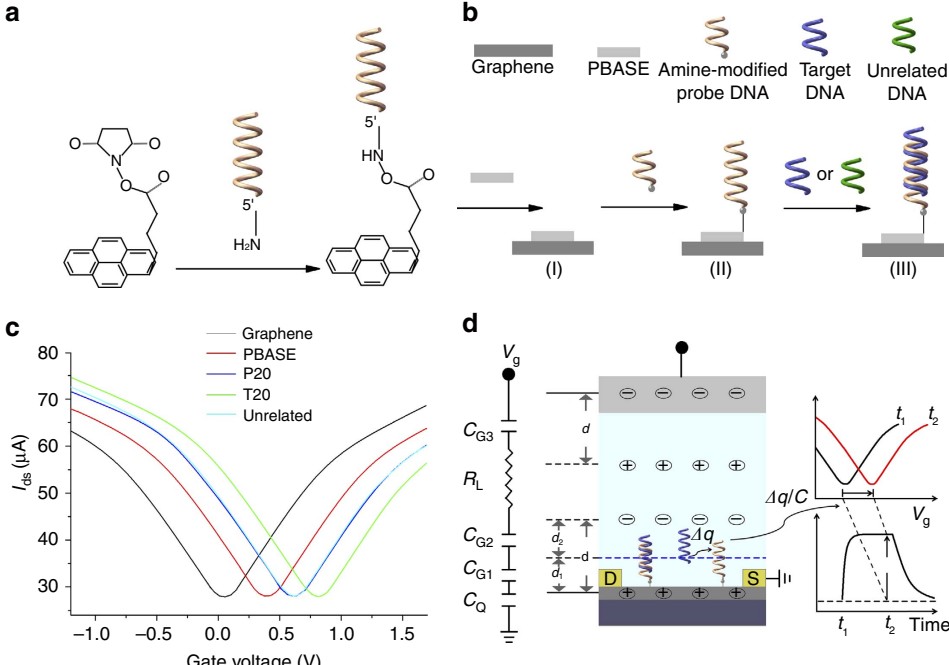

**Figure 2 | Functionalization and properties of G-FETs.** (**a**) Molecular geometry of PBASE and its link to the probe DNA. (**b**) Functionalization of graphene surface by PBASE, immobilization of a probe DNA (in orange) by reacting with the PBASE and hybridization of the probe DNA with target (in blue) and unrelated, control DNAs (in green). (**c**) Source drain current at a constant source drain voltage (0.1 V) with varying gate voltage for the bare graphene (in black) and after each addition of the following reagents in sequence, 10 mM PBASE (in red), 100 nM probe DNA P20 (in blue), 5 nM complementary DNA T20 (in green) and 5 nM unrelated, control DNA U20 (in cyan, essentially indistinguishable from the blue curve). (**d**) A schematic diagram of the sensing model of a G-FET together with the equivalent circuit with four parallel plate capacitors ($C_{G1}$, $C_{G2}$, $C_{G3}$ and $C_Q$) and a resistance ($R_L$) connected in series. $C_{G1}$, $C_{G2}$ and $C_{G3}$ denote the capacitance between graphene and solution, the capacitance of the DNA to solution and the capacitance between Pt gate and solution, respectively. $C_Q$ denotes the quantum capacitance of graphene associated with the finite density of states due to Pauli principle. $R_L$ is the electrical resistance of the ionic solution. When target DNAs dock on the graphene surface, a transfer curve shift ($\Delta V_{cnp}$) occurs due to changes of charges near the graphene sensor chip surface that is continuously monitored.

**Table 1 | Sequences of immobilized probe and target oligonucleotides employed in this work.**

|  | Name | Sequence |
|---|---|---|
| Immobilized oligonucleotide | P20 | H$_2$N-(CH$_2$)$_6$-5'-GAGTTGCTACAGACCTTCGT-3' |
|  | P26 | H$_2$N-(CH$_2$)$_6$-5'-ACCAGGCGGCCGCACACGTCCTCCAT-3' |
|  | P23 | H$_2$N-(CH$_2$)$_6$-5'-TTCAGGCGGCCGCACACGTCCTCCA-3' |
|  | P19 | H$_2$N-(CH$_2$)$_6$-5'-TTTTGGCGGCCGCACACGTCCTC-3' |
|  | P15 | H$_2$N-(CH$_2$)$_6$-5'-TTTTTTCGGCCGCACACGTCC-3' |
|  | P11 | H$_2$N-(CH$_2$)$_6$-5'-TTTTTTTTGCCGCACACGT-3' |
|  | P7 | H$_2$N-(CH$_2$)$_6$-5'-TTTTTTTTTTCCACAC-3' |
| Target oligonucleotides | T26 | 3'-TGGTCCGCCGGCGTGTGCAGGAGGTA-5' |
|  | T26 (T**C**13) | 3'-TGGTCCGCCGGC**C**GTGCAGGAGGTA-5' |
|  | T26 (T**G**13) | 3'-TGGTCCGCCGGC**G**GTGCAGGAGGTA-5' |
|  | T20 | 3'-CTCAACGATGTCTGGAAGCA-5' |
|  | T20 (T**C**01) | 3'-CTCAACGATGTCTGGAAGC**C**-5' |
|  | T20 (T**C**04) | 3'-CTCAACGATGTCTGGA**C**GCA-5' |
|  | T20 (T**C**13) | 3'-CTCAACG**C**TGTCTGGAAGCA-5' |
|  | T20 (T**C**17) | 3'-CTC**C**ACGATGTCTGGAAGCA-5' |
|  | U20 | 3'-ACATGTAGGTTTGATATGAT-5' |

The stability and reproducibility of the $V_g - I_{ds}$ curve was further confirmed by the overlap of forward $V_g$ (red) and backward $V_g$ (green) scanning curves in consecutive sweeps (Supplementary Fig. 4). We also examined the effect of possible current leakage from the Pt electrode on the $V_g - I_{ds}$ curve. The leakage current $I_{gs}$ was measured in buffer with or without DNA as $V_g$ sweeps from − 1.2 to 1.7 V. The leakage current $I_{gs}$ remained smaller than 5 nA and thus is negligible, as the magnitude of the drain source current $I_{ds}$ is at the μA scale (Supplementary Fig. 5).

Figure 2c shows $V_g - I_{ds}$ curves of G-FETs with their minima (charge neutrality point voltages ($V_{cnp}$)) shifting in the positive gate voltage direction after introducing PBASE, probe DNA and complementary DNA, sequentially. The shift in the positive direction by PBASE was explained by its p-doping effect through the charge transfer between the pyrene group and graphene[44], which was also confirmed here by ultraviolet photoelectron spectroscopy, as the work function is increased by ~0.3 eV (Supplementary Fig. 6). The neutrality point voltage $V_{cnp}$ further shifted to the positive by ~0.220 V with the probe DNA (P20), and another ~0.203 V with the target DNA (T20). In contrast, the non-complementary negative control (U20) displays little changes in the $V_g - I_{ds}$ curve, indicating that the G-FET sensors are highly specific. Furthermore, we measured the $V_g - I_{ds}$ curve of G-FETs by addition of other probe DNAs or target DNAs and found that $V_{cnp}$ shifts are all positive in all cases (Supplementary Fig. 7). Positive $V_{cnp}$ shifts induced by DNA are consistent with the previous reports[45–47].

Application of a gate voltage $V_g$ between gate electrode and graphene in solution will lead to the formation of electrical double layers (EDLs) at a polarizable electrode/electrolyte interface. As shown in Fig. 2d. there are two EDLs formed at the Pt-solution and graphene-solution interfaces due to Debye screening[48]. The EDLs at the interfaces can be considered as insulating layer[48,49]. When analytes (target DNA) dock on the surface of the transistor channel, the total gate capacitance ($C$) of a G-FET is made of four parallel plate capacitors ($C_{G1}$, $C_{G2}$, $C_{G3}$ and $C_Q$) connected in series[50] (Fig. 2d) with

$$C = \left( \frac{1}{C_{G1}} + \frac{1}{C_{G2}} + \frac{1}{C_{G3}} + \frac{1}{C_Q} \right)^{-1} \qquad (1)$$

where $C_{G1}$, $C_{G2}$ and $C_{G3}$ denote the capacitance between graphene and solution, the capacitance of the DNA to solution and the capacitance between Pt electrode and solution, respectively. These capacitors are all formed due to EDLs on the interfaces and, thus, called the 'geometrical' capacitances of

the device. $d_1$, $d_2$ and $d$ represent the plate distances of $C_{G1}$, $C_{G2}$ and $C_{G3}$, respectively. $C_Q$ denotes the quantum capacitance of graphene associated with the finite density of states due to the Pauli principle[51]. When the DNA hybridization occurs on the surface of the transistor channel, the additional DNA gives rise to changes in charges ($\Delta q$) at the solution–graphene interface that produce variations in electrostatic potential in graphene channel and positive shifts $V_{cnp}$ by:

$$\Delta V_{cnp} = \frac{\Delta q}{C}. \qquad (2)$$

Here $C$ is the total gate capacitance of the G-FET. Importantly, equation (2) directly defines a relation between the sensor output $\Delta V_{cnp}$ with charge changes $\Delta q$ on the device surface. In addition, from equation (2), the DNA probe density and the hybridization efficiency of duplex formation can also be roughly estimated. For example, the probe density of P20 is estimated to be ~$1.14 \times 10^{11}$ cm$^{-2}$ and its hybridization efficiency to T20 is estimated to be ~92.3% (more details can be found in Supplementary Methods and Supplementary Fig. 8).

**Kinetics and affinity measurements using oligonucleotides**. The above-obtained transfer curves in Fig. 2c and Supplementary Fig. 7 indicated that the device transconductance $g_m$ ($g_m = \partial I_{ds}/\partial V_g$) does not change after the each functionalization step on the graphene surface. As a result, the transfer curve shift (or $\Delta V_{cnp}$) can be determined more simply by fixing $V_g$, detecting the change in the drain-source current $\Delta I_{ds}$ and employing $\Delta V_{cnp} = \Delta I_{ds}/g_m$, as illustrated in Supplementary Fig. 9. This method was used successfully in previously studies[18,38]. Here, each point of $I_{ds}$ is applied for at least 2 s (much longer than the ms relaxation time, Supplementary Fig. 3b) to ensure the reliability of $\Delta V_{cnp}$. The binding kinetics constants of DNA hybridization can be obtained by monitoring the real-time dependence of $\Delta V_{cnp}$ during DNA hybridization on the sensor surface.

Figure 3a shows the kinetics of DNA hybridization at different concentrations of target DNA in one of the six channels (channel 1). Similar kinetic behaviour was observed in all other channels (Supplementary Fig. 10). After injecting target DNAs, $\Delta V_{cnp}$ arises initially and reaches a plateau afterwards. After reaching the plateau, the solution was replaced by a pure 0.01 × phosphate-buffered saline (PBS) buffer to dissociate and remove target DNAs. This leads to a time-dependent decay of $\Delta V_{cnp}$. If an unrelated, control DNA is injected, there is little change in $V_{cnp}$ (open circles in Fig. 3a). Figure 3b shows typical binding

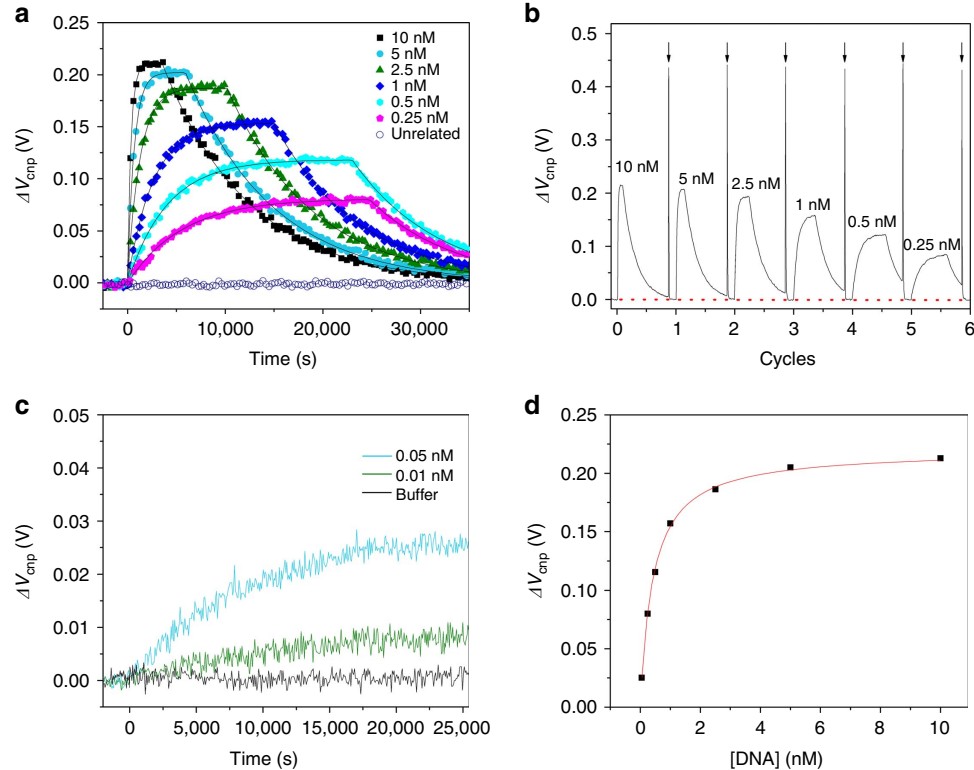

**Figure 3 | DNA–DNA binding using the G-FET in 0.01× PBS buffer.** (**a**) Real-time sensor responses of G-FET to DNA hybridization and dissociation. Each curve represents the measurement of a different T20 DNA concentration from channel 1. The kinetic data for a DNA sequence unrelated to the probe DNA P20 at 5 nM is shown as a control (blue open circles). (**b**) Multiple hybridization cycles upon exposure to the target DNA sequences of T20 at different concentrations. The arrows indicate the timing for the injection of 10 mM NaOH and subsequent rinse by the pure 0.01× PBS buffer. The initial baseline current of the functionalized G-FET is marked by a red dotted line. (**c**) Kinetic curves of the DNA hybridization at 0.05 and 0.01 nM of T20. (**d**) The maximal sensor response $\Delta V_{cnp}$ as a function of DNA concentration in channel 1 along with the curve fitting by equation (10).

cycles using the same G-FET channel. Each dissociation is ended with a fast 60 s pulse injection of 10 mM NaOH/water solution, followed by rinsing with buffer solution between trials. The successful recovery of the original $V_{cnp}$ level at the end of each binding cycle (marked by red dotted line) indicates a complete regeneration for the next binding experiment. That is, the PBASE with attached DNA probes was not removed from the surface and the DNA probe density did not change. This is further supported by obtaining essentially the same kinetic constants from sequential experiments at different DNA concentrations. Thus, the G-FETs are highly stable and reusable.

The kinetics of binding can be fitted by the Langmuir model as in SPR experiments at high flow speeds[6]. The net rate for the formation of the binding complex AB with an immobilized B during a constant flow of A is given by

$$\frac{d[AB]_t}{dt} = k_a[A]\big([B]_{max} - [AB]_t\big) - k_d[AB]_t \tag{3}$$

and the rate of dissociation after the end of the injection is

$$\frac{d[AB]_t}{dt} = k_d[AB]_t \tag{4}$$

where $[AB]_t$, $[A]$ and $[B]_{max}$ denote surface density of bound analyte molecules, the solution concentration of analyte (a constant under the flow condition) and the maximum number of binding sites available per surface area, respectively, and $k_a$ and $k_d$ are association and dissociation rate constant, respectively. Equations (3) and (4) can be solved analytically as the first-order

absorption (equation (5)) and desorption (equation (6)) below[6]:

$$[AB]_t = \frac{k_a[B]_{max}[A]}{k_a[A] + k_d}\left(1 - e^{-(k_a[A] + k_d)t}\right) \tag{5}$$

$$[AB]_t = \frac{k_a[B]_{max}[A]}{k_a[A] + k_d}e^{-k_d t}. \tag{6}$$

The change of surface charges ($\Delta q$) can be expressed as $q_a S [AB]_t$, where $q_a$ is the electric charge contributed by the unit surface density of the adsorbed DNAs to the sensor chip and $S$ is the graphene area. Then, equation (2) can be written as

$$\Delta V_{cnp} = \frac{\Delta q}{C} = \frac{q_a S[AB]_t}{C}. \tag{7}$$

Substituting equations (5) and (6) into equation (7), respectively, yields

$$\Delta V_{cnp} = \frac{\Delta q}{C} = \frac{q_a S}{C}\frac{k_a[B]_{max}[A]}{k_a[A] + k_d}\left(1 - e^{-(k_a[A] + k_d)t}\right) \tag{8}$$

$$\Delta V_{cnp} = \frac{q_a S}{C}\frac{k_a[B]_{max}[A]}{k_a[A] + k_d}e^{-k_d t}. \tag{9}$$

Equations (8) and (9) allow us to fit time-dependent $V_{cnp}$ curves to obtain $k_a$ and $k_d$. Both association and dissociation phases can be well fitted by mono-exponential curves with $R^2 > 0.995$. In principle, both $k_a$ and $k_d$ can be determined from the association phase. However, accurate measurement of $k_d$ requires a separate dissociation phase because $k_d$ is much smaller than $k_a$. For a simple 1:1 binding, the association constants (binding affinity), $K_A$, can be obtained from kinetic measurements by $K_A = k_a/k_d$.

**Table 2 | Kinetic constants of P20-T20 hybridization measured by G-FET sensors.**

| Channel | 1 | 2 | 3 | 4 | 5 | 6 |
|---|---|---|---|---|---|---|
| $k_a$ ($\times 10^5 \, M^{-1} s^{-1}$) | 2.61 (0.11) | 2.68 (0.12) | 2.36 (0.13) | 2.73 (0.09) | 2.38 (0.20) | 2.70 (0.18) |
| $k_d$ ($\times 10^{-4} \, s^{-1}$) | 1.08 (0.07) | 1.13 (0.04) | 1.02 (0.06) | 1.23 (0.10) | 1.10 (0.08) | 1.15 (0.07) |
| $K_A$ ($\times 10^9 \, M^{-1}$)* | 2.35 | 2.37 | 2.31 | 2.22 | 2.16 | 2.42 |
| $K_A$ ($\times 10^9 \, M^{-1}$)† | 2.37 | 2.30 | 2.26 | 2.23 | 2.11 | 2.39 |

*Calculated by $K_A = k_a/k_d$.
†From fitting the concentration-dependent steady state.

The average values of $k_a$, $k_d$ and $K_A$ determined from six G-FET channels are summarized in Table 2. Small s.d. values between kinetic and equilibrium constants from different DNA concentrations (0.25–10 nM) indicate that the G-FET devices are highly reproducible for oligonucleotide sensing. From the six G-FET channels, the average values of $k_a$ and $k_d$ are $2.58 \times 10^5 \, M^{-1} s^{-1}$ and $1.12 \times 10^{-4} \, s^{-1}$, respectively. These results are within the range of $2.3–3.1 \times 10^5 \, M^{-1} s^{-1}$ for $k_a$ and $1.1–1.4 \times 10^{-4} \, s^{-1}$ for $k_d$ by using the SPR method to detect the same DNA sequence[41], and also comparable to the results measured in free solution (that is, all oligonucleotides are mobile) with $k_a \sim 5.2 \times 10^5 \, M^{-1} s^{-1}$ for hybridization of 22-mer oligonucleotides and $k_a \sim 2.5 \times 10^5 \, M^{-1} s^{-1}$ and $k_d \sim 2.1 \times 10^{-3} \, s^{-1}$ for 17-mer oligonucleotides measured by using fluorescence resonance energy transfer[52,53].

However, there is a possibility that the use of a polarizable Pt electrode affects the binding kinetics because it would also induce voltage change across its own electrical double layer capacitance due to DNA hybridization. Our model suggests that this is not the case because the solution-contacting area of Pt electrode is $\sim 2,000$ times larger than that of graphene and $C_{G3} \gg C_{G1}$ or $C_{G2}$ (Supplementary Methods and Supplementary Fig. 8). In other words, contribution of $C_{G3}$ to the overall capacitance (equation (1)) is negligible, similar to a previous study[54]. To further confirm this, we performed the experiments with the same G-FET devices but using a non-polarizable Ag/AgCl electrode (Supplementary Fig. 11). We obtained the association rate constant, $k_a = 2.53 \times 10^5 \, M^{-1} s^{-1}$, the dissociation rate constant, $k_d = 1.15 \times 10^{-4} \, s^{-1}$ and the association equilibrium constant, $K_A = k_a/k_d = 2.20 \times 10^9 \, M^{-1}$. These results are in excellent agreement with those measured using Pt electrode with the average $k_a = 2.58 \times 10^5 \, M^{-1} s^{-1}$, $k_d = 1.12 \times 10^{-4} \, s^{-1}$ and $K_A = 2.30 \times 10^9 \, M^{-1}$, respectively. On the other hand, if the gate electrode had a comparable area with the sensor channel, using a non-polarizable Ag/AgCl electrode would be necessary.

It should be noted that the standard SPR usually have a LOD above 10 nM for short DNA (15-mers)[6] and 0.5 nM for longer DNA (50-mers)[55]. As shown in Fig. 3a, the time-dependent curve of G-FET sensors at 0.25 nM DNA continues to yield reliable kinetic data. As an initial assessment of the LOD of this G-FET sensor, we measured the kinetic curve of hybridization at 0.05 and 0.01 nM of T20 (Fig. 3c). The solution gave an equibration signal that could be easily resolved above the baseline fluctuation, indicating the LOD at $\sim 10$ pM concentration. Moreover, we found that the duration of the experiments highly depends on the analyte concentration. Approximately 3 min are needed for 100 nM and $<1$ min for 1 μM DNA hybridization kinetics (Supplementary Fig. 12).

As $t \to \infty$, equation (8) becomes:

$$\Delta V_{cnp} = \frac{q_a S}{C} [B]_{max} \times \frac{k_a[A]}{k_a[A] + k_d}$$

or, in term of association constant $K_A$,

$$\Delta V_{cnp} = \frac{q_a S}{C} [B]_{max} \times \frac{K_A[A]}{K_A[A] + 1}. \qquad (10)$$

That is, $K_A$ can also be obtained by fitting the concentration-dependent maximal $\Delta V_{cnp}$ at the steady state. Figure 3d shows the steady-state response with respect to a series of DNA concentrations in the range of 0.25 to 10 nM. The sensor responses show that $\Delta V_{cnp}$ first sharply increases as the concentration of DNA increases and gradually becomes saturated above 6 nM. We found that the data were fit well by equation (10) and yielded $K_A = 2.37 \times 10^9 \, M^{-1}$ from channel 1, nearly the same as $2.35 \times 10^9 \, M^{-1}$ calculated from $k_a/k_d$ from the same channel. Results from other channels are also consistent as shown in Table 2 and Supplementary Fig. 13. In addition, the first portion of equation (10), $q_a S[B]_{max}/C$ gives the maximum sensor response and the second portion $K_A[A]/(K_A[A] + 1)$ represents hybridization efficiency. From the curve fitting in Fig. 3d, the maximum sensor response is $\sim 239$ mV, and the hybridization efficiency is at 95.9%, 92.3%, 85.56%, 70.3%, 54.2% and 37.2% corresponding to the T20 concentration of 10, 5, 2.5, 1, 0.5 and 0.25 nM, respectively.

**Single-base-pair mismatch detection.** We further examined the sensitivity of G-FET sensors with respect to single-base mutations to the target sequence. These single-base mismatches are all due to an A to C mutation but at different positions to the target sequence T20 (T**C**01), T20 (T**C**04), T20(T**C**13) and T20(T**C**17) from the 5' end. Moreover, to facilitate the comparison, their nearest neighbouring base pairs are the same (one A-T pair and one G-C pair) except for the T20 (T**C**01), for which the mismatching base pair is located at the 5' end of the target. Mutation sequences can be found in Table 1.

Figure 4a shows that these single-base mismatches are clearly distinguishable from each other with slower association and faster disassociation rates than the fully complementary sequence (Fig. 4g). The $K_A$ of P20-T20 (T**C**01), P20-T20 (T**C**04), P20-T20 (T**C**13) and P20-T20 (T**C**17) are $1.21 \times 10^9 \pm 0.16$, $7.35 \times 10^8 \pm 0.14$, $4.91 \times 10^8 \pm 0.13$ and $9.19 \times 10^8 \pm 0.17 \, M^{-1}$, respectively. The affinity for duplex formation clearly depends on the position of the mismatching base pair. Compared with $K_A$ of $2.31 \times 10^9 \pm 0.14 \, M^{-1}$ for the complementary P20-T20, the P20-T20 (T**C**13) with the mismatched base close to the centre of the sequence has the largest variation with $K_A$ decreased by $\sim 78.7\%$. The $K_A$ values for P20-T20 (T**C**04) and P20-T20 (T**C**17) with the mismatched bases near either 5' or 3' are reduced by $\sim 68.2$ and $\sim 60.2\%$, respectively. The $K_A$ for P20-T20 (T**C**01) with the mismatched base located at the 5' end of the target sequence is lowered only by 47.6%. This result indicates that the mismatches near the centre will disrupt more interactions than the mismatches near terminal ends. Essentially, the same results were obtained when $K_A$ is measured by fitting the concentration-dependent maximal $V_{cnp}$ shift (Supplementary Fig. 14).

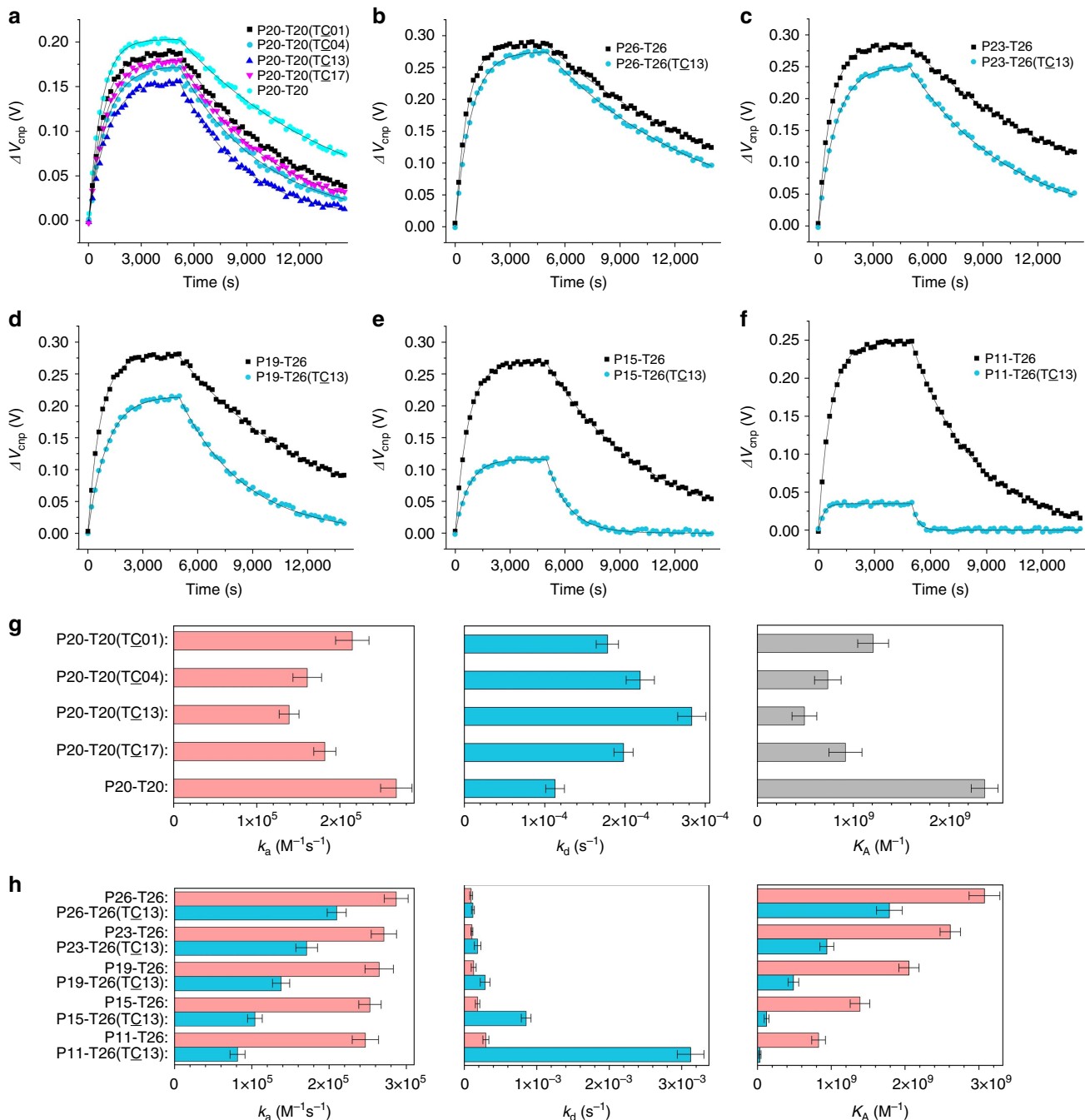

**Figure 4 | Kinetics of DNA hybridization with a single mismatched base pair.** (**a**) Kinetic curves of hybridization of immobilized probe P20 with the complementary T20 and the sequences with one mismatched base pair at different locations. (**b**–**f**) Kinetic curves of hybridization between the designed probes in different lengths from 26 (**b**), 23 (**c**), 19 (**d**), 15 (**e**) to 11 (**f**) and the complementary T26 or between the probe and its mutant T26 (T$\underline{C}$13) as labelled. (**g**) Kinetic rates and association constants from (**a**). (**h**) Kinetic rates and association constants from (**b**–**f**). In all cases, the concentration of the target DNAs is 5 nM. Error bars are s.d.s from measurements of six channels.

We further investigated the DNA sequence T26 whose single-point mutation at the 13th base from the 5′ end (T26(T$\underline{C}$13)) causes Alzheimer's disease[56]. The difference between the time-dependent $V_{cnp}$ curve of P26-T26 and that of P26-T26 (T$\underline{C}$13) is clearly distinguishable (Fig. 4b) with binding affinities at $3.10 \times 10^9 \pm 0.21$ and $1.80 \times 10^9 \pm 0.18\,\text{M}^{-1}$ (last panel in Fig. 4h), respectively. Interestingly, we found that if we shortened the probe sequence to 23, 19, 15, 11 and 7 bases from both ends of P26 but without changing T26, we achieved a significantly larger difference between the time-dependent $V_{cnp}$ curve of Px-T26 and

that of Px-T26 (T$\underline{C}$13) as the probe becomes shorter and shorter (series of panels in Fig. 4b–f for $x = 23$, 19, 15 and 11). When the probe is shortened to 7, there is a clearly visible P7-T26 binding but without any binding signal for the mutant P7-T26 (T$\underline{C}$13) (Supplementary Fig. 15). This indicates that one mismatch is detrimental for the formation of the DNA double strands in 7 bases. The association and dissociate rate constants and binding affinities shown in Fig. 4h indicate that shortening the probe length leads to a small decrease in associate rate constant for the complementary sequence but a much larger reduction for the

mismatched sequence. As a result, the sensitivity to detect the single-base mismatch becomes significantly higher with a shorter probe DNA. The same results are obtained based on the steady-state analysis (Supplementary Fig. 16). Moreover, we further found that G-FETs have the ability to distinguish different mutations in the same position (A→C and A→G at 13th position from the 5′ end, T26(T$\underline{C}$13) and T26(T$\underline{G}$13)) with the probes P15 and P11 (Supplementary Fig. 17 and Supplementary Table 1). The difference between different mutations is large enough for identifying the type of mutation.

## Discussion

We have demonstrated that G-FETs can be used to reliably monitor the kinetics of oligonucleotide binding and unbinding by real-time recording of electric signals, thereby allowing determination of association/dissociation rate constants and equilibrium association constants for DNA–DNA hybridization. These devices are highly specific and capable of discriminating against DNA sequences with single-nucleotide substitutions. Optimal performance of the device in sensitivity, specificity, reliability, reproducibility and reusability requires considering the following factors.

Sensor sensitivity strongly depends on the Debye length determined by the solution ionic strength and temperature[57]. G-FETs can only detect the change of the charge density that occurs within the order of Debye length from the graphene surface. Thus, optimal sensing requires a careful choice of Debye length. As shown in Supplementary Fig. 18, the maximum of the sensor response increases as the ionic strength decreases from $1\times$ PBS to $0.01\times$ PBS buffer and did not change much with further reduction of the ionic strength to $0.005\times$ PBS buffer. The ionic strength of $0.01\times$ PBS buffer yields a Debye length of $\sim 7.3$ nm (ref. 57), which is comparable to the height of the measured DNA binding pair, so that the whole DNA chain can be detected. A buffer with low ionic strength will yield higher sensitivity but at the same time increase nonspecific electrostatic binding of sample molecules to the sensor surface[57]. Thus, unrelated, control DNAs are required to ensure that nonspecific binding at a given ionic strength is not an issue. Here, the high specificity of the sensor for DNA detection can mainly be attributed to the probe-DNA-specific recognition of its complementary partner. In addition, the use of PBASE linkers further reduces nonspecific electrostatic stacking binding of unrelated DNA on the graphene surface[44,46].

One potential source of system errors is the use of the polarizable Pt electrode. We have employed the platinum electrode for FET gating following previous studies (see, for example, ref. 14). The capacitance between the electrode and solution could alter the response time and kinetics. In this study, the device was designed so that the solution-contacting area of the Pt electrode is significantly larger ($\sim 2,000$ times larger) than that of graphene. As a result, the contribution of the Pt-solution capacitance is negligible (equation (1), Supplementary Methods and Supplementary Fig. 8). This is confirmed by the fact that replacing Pt electrode by nonpolarizable Ag/AgCl electrode does not change the kinetics of DNA hybridization measured (Supplementary Fig. 11). In addition, Pt electrode is not functionalized by PBS (Supplementary Fig. 2e) and its current leakage is negligible (Supplementary Fig. 5).

Another important factor affecting sensitivity is the density of probe DNA immobilized on the graphene. Equation (10) shows that the maximal response $\Delta V_{cnp}$ is proportional to the immobilized density of maximum possible DNA probes $[B]_{max}$. In principle, the more DNA probes are immobilized, the higher response level is expected. However, high level of densities of

DNA probes may significantly alter the kinetics of their hybridization with complementary[53]. As shown in Supplementary Fig. 19, higher probe densities ($\geq 6.06\times 10^{11}$ cm$^{-2}$, that is, probes spaced by $\sim 13$ nm) lead to slower target-capture rates. Moreover, kinetic curves obtained at high probe densities ($\geq 6.06\times 10^{11}$ cm$^{-2}$) cannot be fitted well by a single exponential. It is likely that strong interprobe interactions at high densities give rise to unnecessarily complex binding behaviour. Here, we adopted a moderate level of probe density of $\sim 1.14\times 10^{11}$ cm$^{-2}$ (probes spaced by $\sim 30$ nm) for kinetic analysis, at which the LOD reached 10 pM. Although G-FETs have a picomolar LOD, higher analyte concentrations are preferred, if possible, because speedy kinetic measurements can be made at high concentrations, while low DNA concentrations require long time to equilibrate. For example, $\sim 3$ min are needed for 100 nM and $<1$ min for 1 µM DNA hybridization kinetics (Supplementary Fig. 12), but hours at sub-nanomolar concentrations (Fig. 3).

High reproducibility is essential for sensors to be useful. It is affected not only by the cross-channel consistency in device fabrication but also by specific experimental parameters at different steps[58]. Compared with 1D nanomaterials or polycrystalline graphene, graphene single crystal is free from crystal boundary and has low density of dangling bonds on their surfaces and, thus, significantly reduces the electronic scattering and the associated Flicker noise level[38]. This in turn permits low-noise-level detection of DNAs and facilitates consistency of multiple G-FETs in the sensor. Moreover, non-covalent π–π stacking between PBASE linkers and the graphene surface avoids introduction of defects to graphene and retains electrical characteristics of graphene. As shown in Fig. 2c and Supplementary Fig. 7, the slopes of the $I_{ds}-V_g$ curves are essentially identical. We further demonstrated that a fast 60 s pulse injection of 10 mM NaOH solution is sufficient to regenerate the sensor surface so that different concentrations of the analytes can be measured by using the same G-FET device, thus further reducing the errors owing to surface functionalization and device variations. The same sensor chip could be regenerated at least 50 times for multiple measures without significant loss of functionality (recovery $>90\%$), indicative of high reproducibility and reusability. Moreover, the flow rate should be faster than the diffusion rate to ensure steady DNA concentrations available for association, similar to SPR[5,41] or NW affinity sensor[58]. Here we adopted a high rate of 60 µl min$^{-1}$ to ensure sufficiently high mass transport for reliable kinetic measurement.

G-FET biosensors developed here provide a fast, simple and label-free biosensing platform for binding kinetic studies. Owing to their low cost, low power and ease of miniaturization, G-FETs hold great promise for applications where minimizing size, detection time and cost are crucial. Compared with optical sensors, G-FET devices are highly sensitive regardless of the sizes of molecules and have the potential for very large integrated arrays to achieve high-throughput multiplexing assay. Compared with NW sensors, G-FETs are compatible with planar complementary metal-oxide semiconductor fabrication and have the potential to overcome the manufacturing limitation of 1D NW biosensors by costly e-beam lithography. Here, we demonstrated that sensors do not need to be at the nanoscale to achieve pM-level LODs. This means that fabricating G-FETs can achieve a low LOD by using optical lithography rather than exquisite nanolithographic tools. This reduces the time and cost of device fabrication. Moreover, the slow covalent bond formation between the probe DNA and PBASE in G-FETs allows a better control of the probe density to a moderate level by varying the exposure time to the probe solution. This functionalization protocol successfully overcame the issue encountered in NW sensors

where probe DNAs were electrostatically adsorbed onto NW surfaces for immobilization. Such absorption tends to give rise to overly high probe density and in turn lead to 10–100 times smaller association constants of DNA hybridization than that of the common SPR level due to interprobe interactions[14].

In conclusion, we have demonstrated that multi-channel G-FET DNA sensor arrays can capture kinetics of DNA hybridization reproducibly and reliably. An analytical model was developed to estimate probe density, efficiency of hybridization and the maximum sensor response. The device can achieve a detection limit of 10 pM for DNA and distinguish single-base mutations quantitatively in real time. The same sensor chip could be regenerated at least 50 times for multiple measures without significant loss of functionality (recovery > 90%). Owing to the mature FET fabrication technique and simple functionalization, graphene-based sensors will play a significant role in next-generation affinity sensors.

## Methods

**Sensor chip fabrication.** The graphene single-crystal domain with a diameter > 4 mm was grown with CVD by decreasing the graphene nucleation density through oxygen-passivating active sites of Cu surface[59]. That is, we exposed Cu substrates to $O_2$ at a partial pressure of $1 \times 10^{-3}$ torr for 5 min before introducing methane ($CH_4$, $P_{CH_4} = 1 \times 10^{-3}$ torr). Graphene was grown in a quartz tube at 1,050 °C for 12 h. The single-crystal graphene domain was transferred onto a p-doped Si substrate with 300 nm $SiO_2$ layer by the wet transfer method[60]. Then, the graphene was patterned into six separated graphene channels by 60 watts $O_2$ plasma at 200 mTorr for 60–70 s. Then, 20 nm Cr and 100 nm Au layers were thermally evaporated to form the source and drain contacts of each graphene site, with the remaining graphene channel at 45 μm long and 90 μm wide. The metal contacts were annealed in a rapid thermal processor at 450 °C for 1 min to ensure ohmic contacts. To provide room for a 1 cm by 1.5 cm microfluidics chip with a microchannel for solution delivery, the electrical contacts were extended to the edges of the substrate using standard photolithography techniques followed by evaporation of 20 nm Cr and 150 nm Au. To eliminate parasitic current between metal contacts in solution, 80 nm $Si_3N_4$ layer was deposited via plasma-enhanced CVD to passivate these contacts, leaving the graphene area and the outer tips of the Au contacts exposed.

The graphene channels are then packaged with a PMMA microfluidic channel that was fabricated by computer-aided design. The microfluidic channel with a length of 10 mm and width of 0.5 mm was cut in the PMMA layer and two holes with 1 mm diameter were punctured to serve as microchannel inlets and outlets (Fig. 1b). A platinum wire having diameter of 0.5 mm with '⊥' structure is located within the channel right above the devices as a common gate electrode for all the devices (photograph in Fig. 1b and schematic in Supplementary Fig. 8b). The inlet and outlet of the microfluidic channel are connected with tubes for analyte injection and removal. A motorized syringe pump is used for driving the analyte solutions to flow into and out of the microfluidic channel through an inlet/outlet tubing kit. Such a setup enables stable flow of analyte solution and minimizes the noise induced by liquid loading processes, as required for real-time, precise measurement of kinetic processes of DNA hybridization.

**Device functionalization.** The functionalization process was carried out as follows. To immobilize the probe DNA to graphene surface without introducing defects, PBASE was used as a linker[47]. Next, 10 mM PBASE solution in N,N-dimethylformamide was introduced into sensing channels through the inlet tube and to soak graphene for 1 h at room temperature, after which the sensing channels were rinsed three times by the pure N,N-dimethylformamide methanol to wash away any excess reagents. For the immobilization of probe DNAs, single-stranded 1-μM 5′-amine-modified DNA in 1 × PBS buffer flowed through the sensing channel for 1 h. Nonbound DNAs were washed thoroughly with 1 × PBS buffer. Then, 100 mM ethanolamine solution was released into the sensing channel through the inlet tube to deactivate and block the excess reactive groups remaining on the graphene surface. Here 1 × PBS is the standard PBS solution and 0.1 × PBS, 0.05 × PBS, 0.01 × PBS and 0.005 × PBS are diluted from 1 × PBS by ultrapure water. DNA samples were purchased from Sangon Biotech. PBASE and ethanolamine were purchased from Sigma Aldrich. All salts were purchased from Sigma (Sigma Ultra grade) and dissolved in ultrapure water.

**Electronic measurements.** The electrical response $I_{ds}$ of each graphene channel was read by a Keithley 2636 Dual Source Measure Unit. For measurements of transfer characteristics of the G-FETs, $V_{ds}$ was set to 0.1 V and $V_g$ was scanning from −1.2 to 1.7 V. The $V_g$ varied with a sweep step of 50 mV and for each step the given $V_g$ pulse was maintained for 1 s to stabilize $I_{ds}$ to ensure the reliability of $V_g - I_{ds}$ transfer curve. Therefore, the scan rate of $V_g$ in the transfer curve

measurement was 50 mV s$^{-1}$. In the real-time DNA hybridization and dissociation detection, the $\Delta V_{cnp}$ was determined from the ratio of changes in drain source current ($\Delta I_{ds}$) to device transconductance ($g_m$), that is, $\Delta V_{cnp} = \Delta I_{ds}/g_m$ (Supplementary Fig. 9). The change of drain source current was measured in real time by keeping $V_{ds}$ unchanged at 0.1 V and $V_g$ at 0 V (ground potential) to reduce the noise in the system. The $g_m$ was estimated from $I_{ds} - V_g$ measurements for each device using $g_m = \partial I_{ds}/\partial V_g$ without performing actual DNA sensing experiments. Here, each point of $I_{ds}$ was applied for at least 2 s to ensure the stable and reliable measurements. The buffer ionic concentration was optimized to enhance sensitivity (more detailed in Supplementary Fig. 18). Throughout DNA hybridization detecting experiments, 0.01 × PBS buffer (pH 7.8) was used. For kinetic measurement, 'time = 0' was defined as the injection of target DNA. A typical binding cycle includes adsorption, desorption and surface regeneration. The working concentration of each target solution flew at a high rate of 60 μl min$^{-1}$ to ensure sufficiently high mass transport for correct kinetic measurement. After reaching equilibrium, the target solution was exchanged by a pure buffer to dissociate the bound DNA hybrids and to remove the target strands. A 60 s pulse injection of 10 mM NaOH/water solution completely regenerated the probe surface (Fig. 3b). The entire binding cycle was repeated several times at varying concentrations of analyte to generate a robust data set for affinity and kinetics analysis. The G-FETs channels were measured sequentially and were biased at the same time by a common gate electrode (Supplementary Fig. 8b). Only the channel used for measurement passes the current.

**Data availability.** All data are available from the authors on reasonable request.

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

## Acknowledgements

We are grateful for financial support from National Natural Science Foundation of China (11604040, 61671107, 11674199 and 11547225), Shandong Province Natural Science Foundation (ZR2014FQ032), Experimental Technology Program of Dezhou University (SYJS16001, ZZYQ16003) and Taishan Scholars Program of Shandong Province.

## Author contributions

Y.Z., J.W., S.X. and J.Z. conceived the concept. S.X. and J.Z. initiated the sensor design. C.G., Z.L., H.L. and S.X. prepared graphene single-crystal domain. S.G., S.J., S.X., B.M. and W.Y. performed the electrical measurements. Y.Z., J.W., J.Z. and S.X. analysed the data and wrote the manuscript. All the authors edited, discussed and approved the whole paper.

## Additional information

**Competing interests:** The authors declare no competing financial interests.

