## [Peer review file · Nature Communications]

Reviewers' comments:

Reviewer #1 (Remarks to the Author):

The authors present a graphene array of six devices and perform DNA hybridization experiments. They claim a 10 pM sensitivity and present an analytical model explaining their results. The authors use an unconventional platinum electrode to bias the solution which may alter the results on a global array scale.

General Comment

The use of a polarizable platinum reference electrode for gating the solution is very different in comparison to previous solution gated graphene FET works which use a non-polarizable Ag/AgCl reference electrode, e.g. from the references cited [31],[36],[37],[47],[48],[49],[50]. As a platinum electrode mainly operates in the capacitive region without passing Faradiac current, it does a poor job of in controlling the bulk solution at and setting it at a DC voltage which is desired for reliable solution gated graphene FETs. During the voltage sweeps of the IV measurements, the platinum electrode capacitance will charge/discharge, depending upon the swept polarity, and will have a long time constant to return to steady state. Without the scan rate of the voltage sweep from -1.2 V to 1.7 V and further details on measurements, it is hard to say how much this will affect the charge neutrality point measurement. It should most certainly be addressed, considered, and reported, especially in the context of the transient data fittings.

Furthermore, as it is a polarizable electrode similar to graphene FETs used and as it is also exposed to the same analytes as the graphene FETs, it would be assumed that it too will have an induced voltage change across its double layer capacitance due to DNA hybridization. This change of voltage is in series with the solution gating voltage and could therefore produce a false signal across all devices (false meaning not from the same mechanism as reported in the paper). The use of a non-polarizable electrode such as the Ag/AgCl electrode removes this issue and is thus one of the major reasons why it is used in other works. It is this issue that is most pressing and suspect as the 6 devices measure very similar signals (Fig. 3, Supplementary Fig. 5).

In sum, and as discussed below in specific details, if the authors choose to use a platinum electrode to bias the solution, they need to consider and incorporate its effects on the measurements. Performing the experiments with a non-polarizable reference electrode would alleviate this issue and bring the work in accordance with similarly cited previous works.

Specific comments

Fig. 2d, and Page 6 lines 169-193. In considering the effects of DNA charge and the change of potential on the graphene surface, the effects of the capacitance of the DNA to the bulk solution needs to be considered. The authors state the DNA (length of 6.8 nm) resides within a Debye length (7.3 nm) of the graphene; the DNA should also feel a capacitance to the bulk solution which would be added in parallel to the C_c in the calculation of Eq. 1.

As stated in the general comment, the reference electrode capacitance also needs to be considered in series with the gate voltage. As it is functionalized and has a double layer capacitance, similar to the graphene, it too will have a change of voltage due to DNA hybridization which would be recorded by all 6 devices similarly.

Page 15, starting line 511. Are the FETs measured in parallel? It is unclear what “a multiplexed I-V mode” means in terms of the actual measurement. If each device was biased sequentially in time, how were the other devices biased when they were not measured? Did they pass current?

Page 15, starting line 513. What was the scan rate of the gate sweep? Was the gate sweep reversed or returned to zero after measurement? This is especially important for the use of a platinum electrode used as a reference as the RC time constant and charging effects would come into play and affect the transient equations derived.

Was current passing through the reference electrode during I-V measurement? Was a gate current passing through the graphene device under measurement? An I-V plot would be beneficial for the graphene devices in solution with the platinum reference electrode. A major consequence would be electric fields forming in the solution possibly affecting the movement of DNA. Also, where was the reference located within the channel in relation to the devices and inlet/outlet?

Reviewer #2 (Remarks to the Author):

The paper reports on the investigation of DNA-DNA binding kinetics and affinity using graphene field-effect transistor arrays. The authors demonstrate fabrication of 6 G-FET sensors from a centimetre sized single-crystal graphene flakes with very similar sensor characteristics. This fabrication method shows a very promising technological step towards a reliable graphene-based DNA sensing technology that goes well behind single G-FET devices that have been typically reported in the literature so far. The authors employ physisorbed PBASE linkers with chemically attached probe DNA on graphene to detect target DNA on the sensors. This detection approach avoids introduction of defects to graphene, retaining its electrical characteristics and reducing non-specific electrostatic stacking binding of unrelated DNA on the graphene surface. Even though the sensitivity of these sensors is not as high as those using chemically attached DNA probes to graphene, the sensors show excellent reproducibility and recovery. The most interesting part of the paper, however, is the analysis of the real-time G-FET sensor responses to DNA hybridization and dissociation that reveals strong dependence on DNA length and concentration. These experimental findings are modelled by an analytical model to estimate DNA probe density, kinetic rates and association constants. In my view, the presented results are new and interesting and will be useful for researchers working with DNA and graphene bio-sensors. I feel that the manuscript should become acceptable after minor revisions, providing more accurate explanation to the following specific questions and comments.

Specific questions

1. The exact details of the recovery of the sensors are not well explained in the manuscript. It is not clear if the PBASE with attached probe DNA get removed from the surface by the injection of 10 mM NaOH/water solution. There is also no proof that the injection does not change the DNA probe density on the surface.
2. Although the authors claim similarity of the binding kinetic constants determined from SPR and G-FET methods, there is missing a comparison of the DNA binding on functionalized graphene surfaces and in liquids. This is information would be extremely important for readers.
3. As mentioned in the paper the sensor sensitivity strongly depends on the Debye length and gets influenced by the solution ionic strength and temperature. If this is the case then I wonder how it is possible to obtain quantitative analysis of DNA probe density using GFETs for different ionic strengths and DNA concentrations. This question also closely relates to the estimate of the charge of target DNA on the surface for different experimental conditions. Since the mechanism of the DNA detection is based on the DNA charge induced change of electrostatic potential in a graphene channel in liquid, I wonder if the estimate of the charge of DNA is correctly estimated using simplified equation (1) on page 6 (line 189) and one electron charge per nucleotide. This model does not take into account any charge screening and varying distance of different nucleotides in DNA from graphene. In my opinion this can significantly change the quantitative analysis of detected DNA density on the surface.
4. The authors claim that „Equations (5) and (6) show that the sensor response ΔV_{cnp} is independent of the graphene area S .“ I do not think that this can be true also for submicron

graphene devices.

5. There is missing information about the authors and their affiliations in the manuscript.

Reviewers' comments:

Reviewer #1 (Remarks to the Author):

>The authors use an unconventional platinum electrode to bias the solution which may alter the results on a global array scale.

Answer: We employed the platinum electrode for FET gating following previous studies (e.g. *J. Am. Chem. Soc.* 2006, 128, 16323, Ref. [14] and *Biosens. Bioelectron.* 2015, 71, 222). In this revised version, we demonstrated that replacing platinum by a Ag/AgCl electrode does not affect our results. The effect of polarizability in Pt electrode was negligible due to the large difference in solution-contacting areas between Pt electrode and graphene. These results along with other experiments and explanation are provided in detail below.

General Comment

>The use of a polarizable platinum reference electrode for gating the solution is very different in comparison to previous solution gated graphene FET works which use a nonpolarizable Ag/AgCl reference electrode, e.g. from the references cited [31],[36],[37],[47],[48],[49],[50]. As a platinum electrode mainly operates in the capacitive region without passing Faradiac current, it does a poor job of in controlling the bulk solution at and setting it at a DC voltage which is desired for reliable solution gated graphene FETs. During the voltage sweeps of the IV measurements, the platinum electrode capacitance will charge/discharge, depending upon the swept polarity, and will have a long time constant to return to steady state. Without the scan rate of the voltage sweep from -1.2 V to 1.7 V and further details on measurements, it is hard to say how much this will affect the charge neutrality point measurement. It should most certainly be addressed, considered, and reported, especially in the context of the transient data fittings.

Answer: To demonstrate the effect of platinum electrode capacitance on charge/discharge, we showed that the steady state can be reached within a millisecond (~ 0.2 ms in characteristic fall and arise times, Supplementary Fig. S3a and below) in response to a single pulse from 0 to 50 mV. This time scale is three orders of magnitude shorter than 1 second used in each step of V_g pulse (Supplementary Fig. S3b) when the V_g was scanned from -1.2 to 1.7 V with the step of 50 mV per second. These two new figures are discussed in the new revised version (Page 5, last paragraph).

Supplementary Fig. S3. a, Transient response of I_{ds} to V_g pulsed from 0 to 50 mV. **b**, time dependence of I_{ds} as V_g varies from -1.2 to 1.7 V.

>Furthermore, as it is a polarizable electrode similar to graphene FETs used and as it is also exposed to the same analytes as the graphene FETs, it would be assumed that it too will have an induced voltage change across its double layer capacitance due to DNA hybridization. This change of voltage is in series with the solution gating voltage and could therefore produce a false signal across all devices (false meaning not from the same mechanism as reported in the paper). The use of a non-polarizable electrode such as the Ag/AgCl electrode removes this issue and is thus one of the major reasons why it is used in other works. It is this issue that is most pressing and suspect as the 6 devices measure very similar signals (Fig. 3, Supplementary Fig. 5). In sum, and as discussed below in specific details, if the authors choose to use a platinum electrode to bias the solution, they need to consider and incorporate its effects on the measurements. Performing the experiments with a non-polarizable reference electrode would alleviate this issue and bring the work in accordance with similarly cited previous works.

Answer: We performed the experiments with the same G-FET devices but gated by using a non-polarizable Ag/AgCl electrode and demonstrated that the polarizable effect of the Pt electrode does not affect our binding-affinity measurements. We obtained the hybridization rate constant, $k_a = 2.53 \times 10^5 \text{ M}^{-1} \text{ s}^{-1}$, the dissociation rate constant, $k_d = 1.15 \times 10^{-4} \text{ s}^{-1}$ and the association equilibrium constant, $K_A = k_a/k_d = 2.20 \times 10^9 \text{ M}^{-1}$ (Supplementary Fig. S11c). These results are in excellent agreement with those measured using Pt electrode with the average $k_a = 2.58 \times 10^5 \text{ M}^{-1} \text{ s}^{-1}$, $k_d = 1.12 \times 10^{-4} \text{ s}^{-1}$ and $K_A = 2.30 \times 10^9 \text{ M}^{-1}$, respectively, indicating that the voltage change (ΔV_{cmp}) due to DNA hybridization from polarizable Pt electrode is negligible. The negligible polarizable effect of the Pt electrode can be attributed to the fact that the surface area of Pt electrode immersed in the solution ($S_{\text{Pt}} \sim 7.85 \times 10^6 \mu\text{m}^2$) is much larger than that of graphene ($S_{\text{graphene}} \sim 4.05 \times 10^3 \mu\text{m}^2$) and thus the capacitance between the Pt electrode and the solution is much larger than that between the graphene and the solution. As a result, its contribution to the voltage change (ΔV_{cmp}) during DNA hybridization is negligible. More detailed explanations can be seen in the **Response to the Specific comment 1** below. We agree with the reviewer that the effect of polarizability of the Pt electrode would be

non-negligible if the area of the Pt electrode were comparable to that of graphene. In this case, the use of Ag/AgCl electrode would be necessary. These results and discussion were added in paragraphs on Pages 9 (Results) and 14 (Discussion) along with new supplementary figures.

Supplementary Fig. S11c. Real-time sensor responses to DNA hybridization and dissociation using the G-FET gated by using a Ag/AgCl electrode.

Specific comments

>1. Fig. 2d, and Page 6 lines 169-193. In considering the effects of DNA charge and the change of potential on the graphene surface, the effects of the capacitance of the DNA to the bulk solution needs to be considered. The authors state the DNA (length of 6.8 nm) resides within a Debye length (7.3 nm) of the graphene; the DNA should also feel a capacitance to the bulk solution which would be added in parallel to the C_c in the calculation of Eq. 1.

Answer: We are grateful for the reviewer’s suggestion so that we can construct a more complete sensing model. The new model accounts for the capacitance effects of the DNA to the bulk solution as well as the quantum capacitance of graphene. As shown in the new schematic cross-section of G-FET and its equivalent electrical circuit (new Supplementary Fig. S8), the gate capacitance of a G-FET consists of four parallel plate capacitors (C_{G1} , C_{G2} , C_{G3} , and C_Q) connected in series. C_{G1} , C_{G2} , and C_{G3} denote the capacitance between graphene and solution, the capacitance of the DNA to solution, and the capacitance between Pt gate and solution, respectively. C_Q denotes the quantum capacitance of graphene associated with the finite density of states due to Pauli principle. Therefore, the total gate capacitance C is given by

$$C = \left(\frac{1}{C_{G1}} + \frac{1}{C_{G2}} + \frac{1}{C_{G3}} + \frac{1}{C_Q} \right)^{-1}$$

When analytes (target DNAs) dock on the surface of the transistor channel, the additional DNAs give rise to changes in charges (Δq) at the solution-graphene interface. These capacitors will produce variations in electrostatic potential and in turn shift V_{otp} by

$$\Delta V_{cnp} = \frac{\Delta q}{C} = \left(\frac{1}{C_{G1}} + \frac{1}{C_{G2}} + \frac{1}{C_{G3}} + \frac{1}{C_Q} \right) \Delta q = \frac{\Delta q}{C_{G1}} + \frac{\Delta q}{C_{G2}} + \frac{\Delta q}{C_{G3}} + \frac{\Delta q}{C_Q} \quad (R1)$$

The plate distance can be approximated by the Debye length that is theoretically given by $d = 2ce^2/\epsilon_0\epsilon_r k_B T$, where T is the temperature, k_B is Boltzmann's constant and c is the concentration of ions in the electrolyte. The Debye length is calculated to be ~ 7.3 nm in $0.01 \times \text{PBS}$, which can be approximated as the plate distance of C_{G3} . The average plate distance of C_{G1} (d_1) can be approximated as half height of the measured DNA pair. The plate distance of C_{G2} (d_2) can be approximated as Debye length subtracted by d_1 . From the model of the parallel plate capacitors, $C_{G1} = S_{\text{graphene}} \epsilon_r \epsilon_0 / d_1$, $C_{G2} = S_{\text{graphene}} \epsilon_r \epsilon_0 / d_2$, and $C_{G3} = S_{\text{Pt}} \epsilon_r \epsilon_0 / d$, where S_{Pt} is the contact area between the electrolyte and the Pt electrode and S_{graphene} is the contact area between the electrolyte and graphene monolayer. Since $S_{\text{Pt}} (\sim 7.85 \times 10^6 \mu\text{m}^2) \gg S_{\text{graphene}} (\sim 4.05 \times 10^3 \mu\text{m}^2)$ (see the caption in Supplementary Fig. S8 that is reproduced below), the third item $\Delta q/C_{G3}$ in Eq. (R1) (ΔV_{cnp} due to DNA hybridization from Pt electrode) is negligible, as in a previous study (*Phys. Rev. B* 2015, 91, 205413 and *P. Natl. Acad. Sci. USA*, 2011, 108, 13002). The above context is included in Pages 6-7 and Supplementary Fig. S8.

Supplementary Fig. S8. a, A schematic diagram of the sensing model together with the equivalent circuit with four parallel plate capacitors (C_{G1} , C_{G2} , C_{G3} , C_Q) and a resistance (R_L) connected in series. C_{G1} , C_{G2} , and C_{G3} denote the capacitance between graphene and solution, the capacitance of the DNA to solution, and the capacitance between Pt gate and solution, respectively. C_Q denotes the quantum capacitance of graphene associated with the finite density of states due to Pauli principle. R_L is the electrical resistance of the ionic solution. **b**, Schematic diagram of the location of Pt electrode within the channel in relation to the devices and inlet/outlet. The area of Pt (S_{Pt}) immersed in the buffer with a conservative estimate of $\sim 7.85 \times 10^6 \mu\text{m}^2$ (defined by $\pi r \times L = 3.14 \times 0.25 \times 10 \times 10^6 \mu\text{m}^2$, here, r is the radius of Pt wire, L is the length of the microfluidic channel; half of the Pt wire was immersed in the buffer), which is nearly 2000 times larger than the area of graphene S_{graphene} of $4.05 \times 10^3 \mu\text{m}^2$ defined by the graphene channel length of $45 \mu\text{m}$ and width of $90 \mu\text{m}$.

>2. As stated in the general comment, the reference electrode capacitance also needs to be considered in series with the gate voltage. As it is functionalized and has a double layer capacitance, similar to the graphene, it too will have a change of voltage due to DNA

hybridization which would be recorded by all 6 devices similarly.

Answer: As discussed above, the reference electrode capacitance is now considered in the revised sensing model (Pages 6-7 and Supplementary Fig. S8 in the revised version). The voltage change (ΔV_{cnp}) due to DNA hybridization from Pt electrode is negligible compared to that from graphene channel due to large difference in the area immersed in solution. Moreover, the Pt was not functionalized (i.e. not bound with PBASE or DNA probe), as characterized by Raman spectrum (New Supplementary Fig. S2e). That is, there is no direct impact of the Pt electrode on DNA binding. New results are presented in Page 5 and discussed in Page 14.

>3. Page 15, starting line 511. Are the FETs measured in parallel? It is unclear what “a multiplexed I-V mode” means in terms of the actual measurement. If each device was biased sequentially in time, how were the other devices biased when they were not measured? Did they pass current?

Answer: The G-FETs were measured sequentially and were biased at the same time because they have a common gate electrode (Fig. S8b). Only the channel in measurement passes the current. This context is now included in the method section (Page 18).

>4. Page 15, starting line 513. What was the scan rate of the gate sweep? Was the gate sweep reversed or returned to zero after measurement? This is especially important for the use of a platinum electrode used a reference as the RC time constant and charging effects would come into play and affect the transient equations derived.

Answer: The scan rate was 50 mV/s as described in the answer to the general comments. For the real-time DNA hybridization sensing (measuring kinetics and affinity constants), the sensor response ΔV_{cnp} was not obtained by V_g sweeping but by detecting the change in the drain-source current (ΔI_{ds}) due to DNA hybridization at a fixed V_g . This simpler method, successfully employed in previous studies (*Nat. nanotechnol.* 2012, 7, 401 and *Sci. Rep.* 2014, 5, 10546), was possible because transconductance g_m ($g_m = \partial I_{\text{ds}} / \partial V_g$) does not change in the process of functionalization or in the actual DNA sensing, ΔV_{cnp} can be calculated from ΔI_{ds} divided by the device transconductance g_m (i.e. $\Delta V_{\text{cnp}} = \Delta I_{\text{ds}} / g_m$). An explanatory plot is shown in Supplementary Fig. S9 (described in Pages 7 and 17). In addition, our measurement of the V_g - I_{ds} curve is very stable, regardless of forward or backward sweeping as shown in Supplementary Fig. S4 (described in Page 6).

Supplementary Fig. S9. Explanatory diagram for determining the shift of I_{ds} - V_g transfer curve (or ΔV_{cnp}) by $\Delta I_{ds}/g_m$.

Supplementary Fig. S4. V_g - I_{ds} transfer curve of the G-FETs with forward V_g (red) and backward V_g (green) sweeping in consecutive sweeps.

>5. Was current passing through the reference electrode during I-V measurement? Was a gate current passing through the graphene device under measurement? An I-V plot would be beneficial for the graphene devices in solution with the platinum reference electrode. A major consequence would be electric fields forming in the solution possibly affecting the movement of DNA. Also, where was the reference located within the channel in relation to the devices and inlet/outlet?

Answer: The leakage current I_{gs} passing through the reference Pt electrode was measured with V_g sweeping from -1.2 to 1.7 V in buffer and in buffer with DNA. In both cases, the leakage current I_{gs} remained smaller than 5 nA when V_g varied from -1.2 to 1.7 V (Supplementary Fig. S5). In the real-time DNA sensing measurement (kinetic constant

measurement), the V_g was kept at 0 V (ground potential) and at this point I_{gs} is ~ 0.3 nA, which is negligible when compared to the drain source current I_{ds} (> 27 μ A). Because V_g was fixed to 0 V in the actual DNA sensing, there is no gating electric field to affect the movement of DNA. The above context is now included in Pages 6 and 14. The Pt electrode with “ \perp ” structure was located within the channel right above the devices and in the middle between inlet and outlet as shown in Supplementary Fig. S8b, Fig.1b and 1c (now described in Page 16 and the caption of Supplementary Fig. S8b).

Supplementary Fig. S5. Leakage current I_{gs} measured in buffer and in buffer with DNA as V_g sweeps from -1.2 to 1.7 V.

Reviewer #2 (Remarks to the Author):

The paper reports on the investigation of DNA-DNA binding kinetics and affinity using graphene field-effect transistor arrays. The authors demonstrate fabrication of 6 G-FET sensors from a centimetre sized single-crystal graphene flakes with very similar sensor characteristics. This fabrication method shows a very promising technological step towards a reliable graphene-based DNA sensing technology that goes well behind single G-FET devices that have been typically reported in the literature so far. The authors employ physisorbed PBASE linkers with chemically attached probe DNA on graphene to detect target DNA on the sensors. This detection approach avoids introduction of defects to graphene, retaining its electrical characteristics and reducing non-specific electrostatic stacking binding of unrelated DNA on the graphene surface. Even though the sensitivity of these sensors is not as high as those using chemically attached DNA probes to graphene, the sensors show excellent reproducibility and recovery. The most interesting part of the paper, however, is the analysis of the real-time G-FET sensor responses to DNA hybridization and dissociation that reveals strong dependence on DNA length and concentration. These experimental findings are modelled by an analytical model to estimate DNA probe density, kinetic rates and association constants. In my view, the presented results are new and interesting and will be useful for researchers working with DNA and graphene bio-sensors. I feel that the manuscript should become acceptable after minor revisions, providing more accurate explanation to the

following specific questions and comments.

Answer: Thank the reviewer for the thoughtful comments and we further addressed specific questions below.

Specific questions

>1. The exact details of the recovery of the sensors are not well explained in the manuscript. It is not clear if the PBASE with attached probe DNA get removed from the surface by the injection of 10 mM NaOH/water solution. There is also no proof that the injection does not change the DNA probe density on the surface.

Answer. Fig. 3b indicated the typical binding cycles using the same G-FET. After a 60 s pulse injection of 10 mM NaOH/water solution, followed by rinsing with the buffer solution, the initial baseline current of functionalized G-FET was completely restored (red dot line added in the revised manuscript for comparison between original and new baselines), suggesting that the PBASE with attached DNA probes were not removed from the surface and the DNA probe density did not change. The successful recovery of the original V_{cnp} level of functionalized G-FET at the end of each binding cycle indicates a complete regeneration for the next binding experiment. That is, the G-FETs are highly stable and reusable. The complete regeneration of graphene surface with PBS and DNA probes attached is supported by obtaining essentially the same kinetic constants from sequential experiments at different DNA concentrations. A red dotted is now added to indicate the initial baseline current of functionalized G-FET in the revised Fig. 3 to illustrate this point more clearly (Page 8).

>2. Although the authors claim similarity of the binding kinetic constants determined from SPR and G-FET methods, there is missing a comparison of the DNA binding on functionalized graphene surfaces and in liquids. This is information would be extremely important for readers.

Answer: We added a comparison of hybridization kinetics constants measured in free solution (all oligonucleotides are mobile) to the results measured in this work. The manuscript is revised as follows (Page 9):

From the six G-FET channels, the average values of k_a and k_d are $2.58 \times 10^5 \text{ M}^{-1} \text{ s}^{-1}$ and $1.12 \times 10^{-4} \text{ s}^{-1}$, respectively. These results are within the range of $2.3\text{-}3.1 \times 10^5 \text{ M}^{-1} \text{ s}^{-1}$ for k_a and $1.1\text{-}1.4 \times 10^{-4} \text{ s}^{-1}$ for k_d by using the SPR method to detect the same DNA sequence⁴¹, and also comparable to the results measured in free solution (all oligonucleotides are mobile) with $k_a \sim 5.2 \times 10^5 \text{ M}^{-1} \text{ s}^{-1}$ for hybridization of 22-mer oligonucleotides and $k_a \sim 2.5 \times 10^5 \text{ M}^{-1} \text{ s}^{-1}$, $k_d \sim 2.1 \times 10^{-3} \text{ s}^{-1}$ for 17-mer oligonucleotides measured by using fluorescence resonance energy transfer (FRET)^{52,53}

>3. As mentioned in the paper the sensor sensitivity strongly depends on the Debye length and gets influenced by the solution ionic strength and temperature. If this is the case then I

wonder how it is possible to obtain quantitative analysis of DNA probe density using GFETs for different ionic strengths and DNA concentrations. This question also closely relates to the estimate of the charge of target DNA on the surface for different experimental conditions. Since the mechanism of the DNA detection is based on the DNA charge induced change of electrostatic potential in a graphene channel in liquid, I wonder if the estimate of the charge of DNA is correctly estimated using simplified equation (1) on page 6 (line 189) and one electron charge per nucleotide. This model does not take into account any charge screening and varying distance of different nucleotides in DNA from graphene. In my opinion this can significantly change the quantitative analysis of detected DNA density on the surface.

Answer: In DNA binding sensing, the measurements were performed in the same ionic strengths of $0.01 \times$ PBS (in Methods) and the DNA probe density was analyzed in this ionic strength. The DNA concentrations measured are in the level of nM (0.01-10 nM), which is far lower than that in $0.01 \times$ PBS (with 1.37 mM NaCl), thus is negligible in the total ionic strength. Because the height of P20 at ~ 6.8 nm (~ 0.34 nm/nt) is shorter than the Debye length of ~ 7.3 nm in $0.01 \times$ PBS, DNA charges were not screened in the ionic solution for estimation of the DNA probe density of P20. If the distance of DNA to graphene were longer than the Debye length, the effect of charge screening would necessarily be included. To estimate the DNA probe density more precisely, we modified the sensing model by including the capacitance of the DNA to solution and quantum capacitance of graphene, as described in detail in the response to Specific comment 1 to Reviewer #1. Specifically, the ΔV_{cnp} induced by amount of surface charges Δq due to DNA hybridization is given by

$$\Delta V_{cnp} = \frac{\Delta q}{C} = \left(\frac{1}{C_{G1}} + \frac{1}{C_{G2}} + \frac{1}{C_{G3}} + \frac{1}{C_Q} \right) \Delta q = \frac{\Delta q}{C_{G1}} + \frac{\Delta q}{C_{G2}} + \frac{\Delta q}{C_{G3}} + \frac{\Delta q}{C_Q} \quad (R1)$$

Here, C_{G1} , C_{G2} , and C_{G3} denote the capacitance between graphene and solution, the capacitance of the DNA to solution, and the capacitance between Pt gate and solution, respectively. These capacitors are all formed due to EDLs on the interfaces and, thus, called the "geometrical" capacitances of the device. C_Q denotes the quantum capacitance of graphene associated with the finite density of states due to Pauli principle. C is the total gate capacitance of C_{G1} , C_{G2} , C_{G3} , and C_Q connected in series. As parallel plate capacitors, the capacitance is expressed as $C_{G1} = S_{\text{graphene}} \epsilon_r \epsilon_0 / d_1$, $C_{G2} = S_{\text{graphene}} \epsilon_r \epsilon_0 / d_2$, and $C_{G3} = S_{\text{Pt}} \epsilon_r \epsilon_0 / d$, where ϵ_0 , ϵ_r and S_{graphene} are vacuum permittivity (8.85×10^{-12} F/m), the relative dielectric constant of water ($\epsilon_r = 80$), and graphene channel area, respectively. Since $S_{\text{Pt}} \gg S_{\text{graphene}}$, the ΔV_{cnp} due to DNA hybridization from Pt electrode ($\Delta q / C_{G3}$) is neglected. The average distance d_1 between DNA and graphene can be approximated as ~ 3.4 nm by using half height of the measured DNA pair (~ 6.8 nm for P20). The plate distance of DNA to solution, d_2 is estimated to be ~ 3.9 nm (Debye length of ~ 7.3 nm subtracted by d_1). Thus, the total geometrical capacitance (C_{TG}) is estimated of $\sim 3.9 \times 10^{-4}$ μF by

$$\frac{1}{C_{TG}} = \frac{1}{C_{G1}} + \frac{1}{C_{G2}}$$

The C_Q of the graphene channel is estimated of $\sim 8.1 \times 10^{-5}$ μF by $C_Q S_{\text{graphene}}$. Here, C_q is quantum capacitance per unit area of ~ 2 $\mu\text{F cm}^{-2}$ (*Nano Lett.* 2009, 9, 3318 and *Nat.*

Nanotechnol. 2008, 3, 654). C_Q is comparable to C_{TG} and should be taken into account. In the case that the probe DNA length is shorter than the Debye length, the charge changes from T20 with 20 nucleotides can be described as $\Delta q = 20neS_{\text{graphene}}$, where n denote probe density of DNA and e is electron charge. Then, Eq. (R2) can be written as

$$\Delta V_{\text{cnp}} = \frac{\Delta q}{C} = \left(\frac{1}{C_{G1}} + \frac{1}{C_{G2}} + \frac{1}{C_{G3}} + \frac{1}{C_Q} \right) 20neS_{\text{graphene}} \quad (\text{R2})$$

Using the above modified model and ΔV_{cnp} of ~ 0.220 V with P20 addition, we estimated that the probe density (n) of P20 in $0.01 \times$ PBS was $\sim 1.140 \times 10^{11} \text{ cm}^{-2}$. Similarly, the estimated density of the hybridized DNA T20 was $\sim 1.052 \times 10^{11} \text{ cm}^{-2}$ from $\Delta V_{\text{cnp}} = 0.203$ V. Supplementary Fig. S19 compared kinetics of DNA hybridization at different probe densities. The new model is now described in details in Pages 6-7 and Supplementary Fig. S8.

Supplementary Fig. S19.

Supplementary Fig. S19. DNA hybridization kinetics at different probe densities as labeled. DNA probe P20 at different immobilization density hybrids with its complementary DNA T20 at 10 nM concentration. The data shows slower target-capturing rates at higher (4.07×10^{11} and $6.06 \times 10^{11} \text{ cm}^{-2}$) probe densities with poorer exponential fit than at lower (0.72×10^{11} , 1.12×10^{11} and $2.31 \times 10^{11} \text{ cm}^{-2}$) probe densities. Here, all the probe densities were estimated by using the method described in the caption of **Supplementary Fig. S8**.

>4. The authors claim that “Equations (5) and (6) show that the sensor response ΔV_{cnp} is independent of the graphene area S .” I do not think that this can be true also for submicron graphene devices.

Answer: We would like to thank the reviewer for pointing out this and we have modified the sensing model (described in detail in the previous answer). In this new model,

$$\Delta V_{cnp} = \frac{\Delta q}{C} = \left(\frac{1}{C_{G1}} + \frac{1}{C_{G2}} + \frac{1}{C_{G3}} + \frac{1}{C_O} \right) \Delta q$$

Here, $C_{G1} = S_{\text{graphene}} \epsilon_r \epsilon_0 / d_1$, $C_{G2} = S_{\text{graphene}} \epsilon_r \epsilon_0 / d_2$, and $C_{G3} = S_{\text{Pt}} \epsilon_r \epsilon_0 / d$. Assuming q_a is the electric charge contributed by the unit surface density of the adsorbed DNAs to the sensor chip. The change of surface charges (Δq) is expressed as $q_a S_{\text{graphene}} [AB]_t$. In the modified model, as $S_{\text{Pt}} \neq S_{\text{graphene}}$ the graphene area S was retained in the following relationship,

$$\Delta V_{cnp} = \frac{\Delta q}{C} = \frac{q_a S_{\text{graphene}} [AB]_t}{C}$$

where ΔV_{cnp} is independent of S_{graphene} only in the case of $S_{\text{Pt}} \gg S_{\text{graphene}}$ and the contribution of $1/C_O$ to the total capacitance C is negligible. Because $1/C_O$ is not negligible, ΔV_{cnp} is not independent of S_{graphene} as the reviewer suggested. The new model is now described in details in Pages 6-7 and Supplementary Fig. S8.

>5. There is missing information about the authors and their affiliations in the manuscript.

Answer: The authors and their affiliations were removed at the request of the editor.

REVIEWERS' COMMENTS:

Reviewer #1 (Remarks to the Author):

The authors have done a good job with their revision to expand and justify their experiments. They have addressed all of my previous concerns. I would like to recommend publication of this manuscript in the journal.

Reviewer #2 (Remarks to the Author):

All of my concerns and comments have been answered in the rebuttal letter and revised manuscript. I recommend the paper for publication.